# The Cooperative Anti-Neoplastic Activity of Polyphenolic Phytochemicals on Human T-Cell Acute Lymphoblastic Leukemia Cell Line MOLT-4 In Vitro

**DOI:** 10.3390/ijms23094753

**Published:** 2022-04-26

**Authors:** Patrycja Koszałka, Grzegorz Stasiłojć, Natalia Miękus-Purwin, Maciej Niedźwiecki, Maciej Purwin, Szymon Grabowski, Tomasz Bączek

**Affiliations:** 1Institute of Medical Biotechnology and Experimental Oncology, Intercollegiate Faculty of Biotechnology, University of Gdańsk and Medical University of Gdańsk, Dębinki Street 1, 80-211 Gdańsk, Poland; pkosz@gumed.edu.pl (P.K.); gstasilojc@gumed.edu.pl (G.S.); 2Department of Pharmaceutical Chemistry, Medical University of Gdańsk, Hallera Street 107, 80-416 Gdańsk, Poland; miekusn@gmail.com (N.M.-P.); maciejpurwin@o2.pl (M.P.); 3Department of Pediatrics, Hematology and Oncology, Medical University of Gdansk, Debinki Street 7, 80-211 Gdańsk, Poland; maciej.niedzwiecki@gumed.edu.pl; 4GetResponse Cares Foundation, Arkońska Street 6/A3, 80-387 Gdańsk, Poland; simon@getresponse.com

**Keywords:** acute lymphoblastic leukemia, MOLT-4, BJ, curcumin, genistein, resveratrol, quercetin, cell membrane permeability, mitochondrial membrane potential

## Abstract

Acute lymphoblastic leukemia (ALL) is the most common hematological malignancy affecting pediatric patients. ALL treatment regimens with cytostatics manifest substantial toxicity and have reached the maximum of well-tolerated doses. One potential approach for improving treatment efficiency could be supplementation of the current regimen with naturally occurring phytochemicals with anti-cancer properties. Nutraceuticals such as quercetin, curcumin, resveratrol, and genistein have been studied in anti-cancer therapy, but their application is limited by their low bioavailability. However, their cooperative activity could potentially increase their efficiency at low, bioavailable doses. We studied their cooperative effect on the viability of a human ALL MOLT-4 cell line in vitro at the concentration considered to be in the bioavailable range in vivo. To analyze their potential side effect on the viability of non-tumor cells, we evaluated their toxicity on a normal human foreskin fibroblast cell line (BJ). In both cell lines, we also measured specific indicators of cell death, changes in cell membrane permeability (CMP), and mitochondrial membrane potential (MMP). Even at a low bioavailable concentration, genistein and curcumin decreased MOLT-4 viability, and their combination had a significant interactive effect. While resveratrol and quercetin did not affect MOLT-4 viability, together they enhanced the effect of the genistein/curcumin mix, significantly inhibiting MOLT-4 population growth in vitro. Moreover, the analyzed phytochemicals and their combinations did not affect the BJ cell line. In both cell lines, they induced a decrease in MMP and correlating CMP changes, but in non-tumor cells, both metabolic activity and cell membrane continuity were restored in time. (4) Conclusions: The results indicate that the interactive activity of analyzed phytochemicals can induce an anti-cancer effect on ALL cells without a significant effect on non-tumor cells. It implies that the application of the combinations of phytochemicals an anti-cancer treatment supplement could be worth further investigation regardless of their low bioavailability.

## 1. Introduction

Acute lymphoblastic leukemia (ALL) is the most common hematological malignancy in the pediatric patient group. Presently, the cure rate is ~90% due to the use of intensive chemical treatment based on risk-adapted stratification to the appropriate therapeutic subgroups based on the evaluation of minimal residual disease (MRD) and cytogenetic and molecular genetic studies [1,2]. More advanced and sophisticated treatments are continuously being developed, such as the use of protein kinase inhibitors (e.g., imatinib, ruxolitinib), bispecific antibodies (Blinatumomab, Inotuzumab, ozogamicin), and genetically engineered T-cells (CAR-T), which represent a completely new view on the treatment of acute lymphoblastic leukemia [3,4]. However, regardless of the choice of therapeutic method, most patients have numerous early and late complications, especially in intensively treated high-risk groups, such as children with congenital immune deficiencies (Down syndrome and ALL/AML, Nijmegen syndrome, and non-Hodgkin lymphoma) [5].

It is also generally believed that the maximum well-tolerated doses of cytostatics arranged in therapeutic regimens have already been achieved in pediatric hematology [6], which means that increasing the total dose of cytostatics used would be associated with difficult-to-accept side effects or even death. Furthermore, cancer is one of the diseases that occur due to the perturbations of multiple signaling pathways [7]. Therefore, a non-toxic, highly effective, multi-faceted treatment, one that is readily available and cost-effective, is highly desirable.

One of the potential alternatives could be supplementation of the current therapeutical conventional “gold standards” with naturally occurring phytochemicals with chemopreventive properties that inhibit, reverse, or suppress tumorigenesis. Such phytochemicals possess generally lower toxicity versus synthetic therapies and are more acceptable to patients. Among the small molecules derived from fruits and vegetables studied for cancer treatment, polyphenols (quercetin, curcumin, resveratrol, genistein) are pointed out to be effective and safe [8,9,10].

Curcumin is a polyphenolic compound extracted from the rhizome of *Curcuma longa* and related species with wide therapeutic efficacy proven against a broad range of tumors in clinical studies, including liver, lung, and peritoneum cancers [11,12]. It was determined to be non-cytotoxic to normal cells taken orally by cancer patients at 450–3600 mg/day for up to 4 months [11] or even up to 8000 mg/day for 3 months [13]—the dosages required for therapeutic efficacy. It was also well tolerated by healthy volunteers at doses ranging from 500 to 12,000 mg/day for 3 months [14]. Genistein is a hydrolysis product of the 7-O-β-d-glucoside form of genistein naturally occurring in plants. It prevents, delays or blocks multiple steps of carcinogenesis in vitro and in vivo by targeting cellular mechanisms relevant to oxidative stress management, angiogenesis, cell cycle regulation, and apoptosis [15]. It is a phytoestrogen, but it was shown to be a legitimate concern only for Western women diagnosed with breast cancer or at high risk for this cancer [16]. Its high dose (up to 200 µg/mL) induced no damage to normally proliferating lymphocytes in vitro [17], while its oral application in in vivo studies induced no genotoxic [18] and minimally toxic effects [19,20]. Quercetin is one of the most abundant flavonoids present in fruits and vegetables, as well as in wine and tea [21,22]. It was shown in pre- and clinical trials as non-mutagenic and non-toxic to humans when infused intravenously [23,24]. Resveratrol is a non-toxic natural product found mainly in red grapes, grape products, and red wine [25,26]. All these phytochemicals take part in multiple clinical trials against a wide range of neoplasms, with curcumin undergoing even 3rd phase clinical trials for prostate cancer (NCT03769766) [27]. 

In preclinical in vitro studies, they demonstrated high potential as anti-leukemia agents. In a wide range of leukemia cell lines, curcumin can induce epigenetic changes, downregulate DNA methyltransferase I, and inhibit NF-κB activation and downregulation [28,29,30], as it can impact a diverse range of molecular targets and signaling pathways [8]. In the chronic myeloid leukemia cell line, it is able to induce both autophagy and apoptosis via downregulation of the Bcl-2 protein [31]. Genistein and resveratrol induce apoptosis in leukemia cell lines with genistein, which is also able to induce cell cycle arrest and resveratrol to trigger an autophagic death [17,32,33,34,35]. Quercetin inhibits proliferation and induces cell cycle arrest and apoptosis in leukemia cells [36,37,38]. 

However, most of the in vitro data are based on the results obtained when high concentrations of analyzed compounds are used, unachievable in vivo by oral application [39]. Furthermore, many factors make the application of polyphenols in anti-cancer treatment problematic. The utility of curcumin is hindered not only by its low oral bioavailability but also by its poor solubility, rapid metabolism and systemic elimination, even if it is partially ameliorated by its high bioactivity [40,41,42]. Genistein has low solubility and low oral bioavailability due to its rapid metabolization and high activity of efflux transporters [43]. Quercetin is rapidly metabolized, especially when administered intravenously [44]. Resveratrol is also rapidly absorbed and metabolized, with low oral bioavailability [45]. It can also induce contradictory effects depending on its serum concentrations, inducing apoptosis or cell survival [46]. 

Multiple approaches are being sought to overcome these limitations, with limited success [40,41]. The most notable is an application of different oral administration forms, e.g., the changes of polymer-based systems used for the encapsulation of polyphenols to increase their bioavailability and solubility [47,48,49,50]. However, polyphenols were shown to increase their bioactivity and prolong their effectiveness also through their synergistic or additive interactions [8,51]. However, in the case of leukemia cells, there is little information available regarding the anti-tumor effects of their combinations [51].

The main aim of the presented research was to search for the combinations of biologically active substances that can affect the development and progression of ALL at concentrations that were shown to be bioavailable in vivo. We analyzed curcumin, genistein, quercetin, and resveratrol, four dietary polyphenols with the highest anti-leukemia potential demonstrated in the literature. We used a human leukemia MOLT-4 cell line derived from acute lymphoblastic leukemia as an in vitro model for ALL [52] and a normal human foreskin fibroblast cell line (BJ) for toxicity evaluation.

## 2. Results

### 2.1. Genistein and Curcumin Strongly Decrease the Viability of a Human Acute T Lymphoblastic Leukemia MOLT-4 Cell Line

We used a human MOLT-4 cell line derived from acute lymphoblastic leukemia as an in vitro model for ALL [52] to analyze an anti-leukemia potential of curcumin, genistein, quercetin, and resveratrol concentrations that were shown to be bioavailable in vivo. The changes in cell viability were measured with an MTT metabolic assay 24, 48, and 72 h after the application of a single dose of a chosen phytochemical. To compare the effect induced by different phytochemicals on MOLT-4 viability, the concentration range was chosen according to the specific bioavailability of each dietary polyphenol (Table 1).

Curcumin can cause G2/M cell cycle arrest at only 5 μM [28,29,30], but to induce a biologically significant cytotoxic effect on MOLT-4 cells in 48 hours, a concentration high above 10 µg/mL (ab. 27 µM) was indicated [60]. Using low bioavailable doses (6.1 µM, 2.25 µg/mL, 1×), we observed a decrease in MOLT-4 viability (by about 16 ± 5%) after exposure time was prolonged to 72 h (Figure 1). The doubling of its bioavailable dose (12.2 µM, 2×) had a much faster and stronger effect, as viability decreased by 45% (±13%) only after 24 h and remained stable up to 72 h.

In MOLT-4 cell culture, genistein can induce cell cycle arrest in S-phase at 5 µg/mL (ab. 18.5 µM) in only 4 h, but to decrease cell viability by half in 24 h, a concentration of over 13 µg/mL (ab. 48.1 µM) was indicated [17]. We observed a decrease in MOLT-4 viability for a much lower bioavailable dose (11.1 µM, 3 µg/mL, 1×) (by approx. 23 ± 7%) after exposure time was prolonged to 48 h (Figure 1). This effect held stable after 72 h, and was significantly stronger than the effect of curcumin at a comparable dose. The double dose (2×) had a much faster and stronger effect, as viability decreased by 31% (±5%) in only 24 h and by 79% (±8%) after 72 h.

Quercetin inhibits proliferation of leukemia cells at a concentration of ≥50 μM at 48 h [36], induces cell cycle arrest in the late G1 phase of the cell cycle at 70 μM [37], and decreases cell viability through apoptosis at 60–100 μM [36,61]. Using significantly lower bioavailable doses (4.96 µM, 1.5 µg/mL, 1×), we observed no effect on MOLT-4 viability (Figure 1). Its doubling (2×) did not change that. While there are some data showing that a comparable concentration can reduce MOLT-4 viability, it was observed in the presence of 0.2% of DMSO [62,63]. At only 0.1%, DMSO can induce a dose-dependent delay in cell cycle progression as well as significant changes in gene expression [64,65].

Resveratrol inhibits cell viability in a concentration- and time-dependent manner, with a concentration above 10 µM needed to affect MOLT-4 cells [33]. At a lower bioavailable dose (2.2 µM, 0.5 µg/mL, 1×), it did not affect MOLT-4 viability, while a double dose (2×) induced a temporary increase in viability after 24 h (Figure 1). As the MTT assay is based on the activity of mitochondrial enzymes, and resveratrol can improve mitochondrial function [66], this effect was not unexpected.

Therefore, we have demonstrated that genistein and curcumin can decrease MOLT-4 cell line viability at the bioavailable concentration with prolonged exposure, with genistein having a significantly stronger effect (*p* < 0.01) (Figure 1). Resveratrol and quercetin had no significant effect on MOLT-4 viability. 

### 2.2. Genistein and Curcumin Have a Synergistic Effect against a MOLT-4 Tumor Cell Line, Which Can Be Enhanced by the Addition of Quercetin and Resveratrol

As the next step, we analyzed whether a combination of dietary polyphenols can have a synergistic effect in decreasing MOLT-4 cell line viability when combined at a comparable dose according to their bioavailability. 

Curcumin was shown to have synergistic interactions with other nutraceuticals, including genistein, resveratrol, and quercetin [8]. Combined with genistein, it strongly inhibited 17 beta-estradiol-induced proliferation of the human breast cancer cell line MCF-7, when applied at around double the bioavailable dose (10 µM of curcumin and 25 µM of genistein), while single compounds had no significant effect [67]. For the MOLT-4 cell line, we observed that a combination of genistein with curcumin (G/C) at a 1× bioavailable dose (11.1 µM of genistein combined with 6.1 µM of curcumin) induced a significant decrease in cell viability compared to the control (decrease by 11 ± 3%) (Figure 2a) after 24 h, earlier than the effect induced by single compounds. For a 2× dose, a combined effect of curcumin and genistein was comparable to single compounds (Figure 2a). After 48 h, the combined G/C effect at a 1× dose on MOLT-4 was even stronger (a drop by 19%), with significant differences compared to curcumin and genistein alone. For a 2× dose, the effect of curcumin and genistein was synergistic (CI Value 0.77804 as calculated by CompuSyn) for the same incubation period. After 72 h, the effect of genistein and curcumin mix at 1× dose was still holding at a similar level (decrease by 35 ± 2%). After this incubation period, we also observed a decrease in viability induced by 0.5× dose of G/C mix, but at a level comparable to genistein alone, while for a 2× dose of G/C mix, the effect was still stronger than single compounds and still increased compared to the effect after 48 h (a drop by an additional 9%) (Figure 2a).

Quercetin, when mixed with genistein at a 2× dose (9.92 µM of quercetin with 22.2 µM of genistein), had slightly but significantly increased MOLT-4 cell viability after 72 h, reducing the effect of genistein alone (*p* < 0.01) (Figure 2a). When quercetin was combined with resveratrol, it did not affect MOLT-4 viability (Figure 2a).

However, the addition of resveratrol and quercetin to the genistein/curcumin mix at 1× bioavailable dose (11.1 µM of genistein, 6.1 µM of curcumin, 4.96 µM of quercetin, 2.2 µM of resveratrol) (G/C/Q/R) decreased MOLT-4 viability by an additional 16% (*p* < 0.001) compared to G/C mix (Figure 2b). The difference was still present after 48 and 72 h (18%, *p* < 0.001; 12%, *p* < 0.01; respectively). The differences were present also for a higher 2× dose after 24, 48, and 72 (decrease by 32%, 16%, and 10% compared to G/C mix, respectively; *p* < 0.001); however, they also emerged for the lower 0.5× dose after 24 and 48 h (decrease by 9% and 12%, respectively; *p* < 0.001). 

The addition of resveratrol alone to the genistein/curcumin mix had no effect at 1× or 0.5× dose, but for 2× dose, it had increased the viability of MOLT-4 cells after 48 and 72 h of incubation (by approx. 7%, *p* < 0.001 and *p* < 0.01, respectively) (Figure 2b). Therefore, quercetin and resveratrol, while they did not decrease MOLT-4 viability by themselves, were able to enhance the effect induced by the genistein/curcumin mix, indicating the potentiation effect of these two reagents. We did not observe such an effect for curcumin alone (Appendix A), indicating a significant role of genistein in an interactive activity of G/C/Q/R mix.

However, MOLT-4 cell density grows by ab. 70% every 24 h. We analyzed the effect of genistein, genistein/curcumin (G/C), and the mix of all phytochemicals (G/C/Q/R) on the population growth of MOLT-4 cells in vitro. We observed that while a combination of all four polyphenols at a 1× bioavailable dose (6.1 µM curcumin, 11.1 µM genistein, 4.96 µM quercetin, and 2.2 µM resveratrol) was unable to induce an exponential decrease in the cell population over time (it required 2× dose applicable only in vitro) it was able to suppress its growth, especially during the first 48 h of treatment (Figure 3).

### 2.3. Genistein and Its Mixes Do Not Affect the Viability of a Normal Human Foreskin Fibroblast Cell Line BJ

One of the main points in the application of polyphenols in anti-cancer therapy is their well-documented safety. They were demonstrated to be non-toxic to normal cells and well-tolerated in clinical trials in the case of curcumin [11,13,14], quercetin [23,24], and resveratrol [25,26] or, when genistein is concerned, have minimal toxicity when applied at a bioavailable concentration [18,19,20]. However, we still needed to evaluate whether genistein mixed with other nutraceuticals can also affect human cells that did not undergo neoplastic transformation.

To identify chemicals that could cause illness in humans via systemic or local routes after a single exposure, OECD Guidance Document 129 recommends the use of mouse BALB/c 3T3 embryonic fibroblasts or normal human epidermal keratinocyte NHK cells [68]. However, as the population doubling capacity of NHK cells is low, the application of a normal human foreskin fibroblast BJ cell line, with a long lifespan in comparison with other normal human fibroblast cell lines, was validated for toxicity testing [69] according to OECD guidelines. 

Therefore, we used the BJ cell line to test if genistein, its combination with curcumin (G/C) or a mix of all phytochemicals (G/C/Q/R) has any effect on the viability of normal human cells. All the analyzed combinations showed no significant effect on the viability of BJ cells. They have even shown some tendency to increase their viability after 24 h when nutraceuticals were applied at a 2× bioavailable concentration (12.2 µM of curcumin, 22.2 µM of genistein, 9.92 µM of quercetin, and 4.4 µM of resveratrol) (Figure 4). Furthermore, no decline in cell density population was observed in time, with cells reaching a growth plateau after 24 h due to contact inhibition (Figure 5).

### 2.4. Mitochondrial Membrane Potential (MMP) and Cell Membrane Permeability (CMP) after the Curcumin, Genistein, Quercetin, and Resveratrol Mix Treatment

One of the most common ways for nutraceuticals to inhibit the survival of neoplastic cells is by inducing apoptosis [9]. Therefore, to analyze the efficiency of a combination of curcumin, genistein, quercetin, and resveratrol in inducing cancer cell death, we analyzed changes in mitochondrial inner membrane potential, as mitochondrial dysfunction plays a central part in this process. Furthermore, depending on the intensity of the mitochondrial insult, the cell can undergo apoptosis, necrosis, and/or autophagic cell death [70]. Therefore, we also analyzed changes in cell membrane permeability, a basic parameter of necrosis, and later changes in apoptotic cell death, secondary necrosis [71].

A combination of all four analyzed nutraceuticals at 1× bioavailable dose (11.1 µM of genistein, 6.1 µM of curcumin, 4.96 µM of quercetin, 2.2 µM of resveratrol) (G/C/Q/R) decreased MOLT-4 mitochondrial membrane potential (MMP) by about 35 ± 8% compared to the control (*p* < 0.001) after 24 h of incubation (Figure 6a). The decrease has kept stable for the next 48 h. For this combination, half of a bioavailable dose (0.5×) had also some effect (decrease by approx. 10% compared to the control; *p* < 0.001) for the first 48 h after supplementation with the mix. In addition, 2× dose had an effect comparable to that of a mitochondrial oxidative phosphorylation uncoupler, CCCP, used as a positive control (Figure 6a). At 24 h point, the dose-dependent decrease in MMP cells correlated with a dose-dependent increase in the percentage of MOLT-4 cells with a permeable cell membrane (increase by 18 ± 10%, 81 ± 14%; 97 ± 2% compared to the control for 0.5×, 1× and 2× dose, respectively, *p* < 0.001) (Figure 6c). The decrease in CMP at further time-point correlated with an observed decrease in cell viability of MOLT-4 cells (Figure 2b) and an inhibition of the cell population growth (Figure 3) induced by C/G/Q/R mix. Therefore, such reduction is probably not the sign of recovery, but rather a depletion of the population part sensitive to the activity of analyzed polyphenols. Such depletion is most probably the result of mitochondrial injury leading to the disruption of cell membrane continuity, characteristic of necrosis. The observed effect was cytotoxic and not cytostatic as confirmed by cell cycle analysis. No statistically significant differences were observed in the cell cycle between MOLT-4 control cells or cells treated with the analyzed polyphenols (Appendix A).

When a normal human foreskin fibroblast BJ cell line was analyzed, the changes induced by C/G/Q/R mix at 1× bioavailable dose were non-linear, with a sharp drop in MMP after the first 24 h (by 91 ± 0.5% compared to the control; *p* < 0.001), significant recovery after 48 h (by 79 ± 10%; 14% difference with the control) and then some drop to the level observed in MOLT-4 cells after 74 h (by 50 ± 6% compared to the control; *p* < 0.001) (Figure 6b). We observed a similar change for 0.5× dose of the mix, while for 2× dose, MMP decreased to the level of the positive control, CCCP, with the partial recovery of MMP after 72 h (Figure 6b). The changes in CMP were similar to the ones observed for MOLT-4 cells, even if slightly less steep (by approx. 5–9% for 1× dose). However, as this combination of nutraceuticals did not induce any changes in the cell density of the BJ cell population as indicated by the viability assays (Figure 4 and Figure 5), such changes indicate that while this mix induced a significant mitochondrial injury, the metabolic activity of cells was for the most part resumed, and cell membrane continuity restored. Especially when a bioavailable dose was applied (1×).

When the effect of the G/C/Q/R mix was compared with the mix of curcumin and genistein only, the effect on MMP for MOLT-4 and BJ cells was increased by the addition of quercetin and resveratrol. However, when CMP was analyzed, this addition increased the percentage of MOLT-4 cells with permeable cell membrane but had no effect when BJ cells were concerned (Appendix A). This confirms a difference in the changes induced in the CMP of neoplastic and normal cells by the mix of the analyzed phytochemicals. 

## 3. Discussion

The current paradigm for cancer treatment is one of a multi-faceted approach, as cancer is a cytogenetic disease with many metabolic pathways taking part in the regulation of its progression [72]. Nutraceuticals can take part in such multitargeted treatment; however, their application is hindered mainly by their low oral bioavailability [9,39]. Here, we demonstrated that biochemical interactions between the nutraceuticals analyzed in this research allow their anti-cancer activity to be increased against acute lymphoblastic leukemia (ALL) at concentrations that are bioavailable in vitro [33,53,54,55,56,57,58,59]. While genistein had the strongest effect on the viability of the MOLT-4 cell line, stronger than the more popular curcumin, at a bioavailable dose, it was not sufficiently effective to induce a significant inhibitory effect on such a fast-growing cell line. However, by combining genistein with curcumin, we obtained a significantly stronger effect, decreasing MOLT-4 viability as a lower dose (0.5× bioavailable dose) and at an earlier time-point (24 h for 1× dose).

An interactive effect between genistein and curcumin was not surprising, as curcumin has a high potential for synergistic activity [9]. It can target various pathways responsible for the proliferation, the activation of protein kinases, cell survival, tumor suppression, caspase activation, and death receptor activation as well as many others, including transcription factors and epigenetic regulators [8,73,74]. Furthermore, various studies demonstrated that curcumin can induce both autophagy and apoptosis via the regulation of the Akt/mTOR/S6K axis [31,75,76]. Curcumin was shown to inhibit mTORC1 (mammalian target of rapamycin complex 1) signaling even at a low concentration (2.5 µM) as well as phosphorylation of its direct effector, S6 kinase 1 [77,78]. For genistein, the inhibition of S6 kinase (S6K) phosphorylation was pointed out as a mechanism through which it can induce long-term changes in protein synthesis [79,80]. Thus, the increased effect of the curcumin/genistein mix observed here could be due to interactive effects through the regulation of S6 kinase activation resulting in apoptosis via the Akt/mTOR/S6K axis. Both genistein and curcumin were demonstrated to induce mitochondrial dysfunction through mitochondrial permeability transition (MPT) followed by a decrease in MMP [81,82,83]. This well-researched and confirmed mechanism of action for curcumin and genistein can explain the long-term changes in MOLT-4 viability and MMP observed in our research when both phytochemicals are known to be rapidly metabolized [41,42,43]. 

The effect of curcumin/genistein mix could be further enhanced by its supplementation with quercetin and resveratrol, resulting in a stronger effect for 1× dose and an earlier effect for a lower 0.5× dose. The mechanism of action of both these phytochemicals is also quite well researched. Quercetin was shown to induce mitochondria-mediated apoptosis in cancer cells through the inactivation of Akt-1 in the Akt/mTOR axis [84,85,86], a change that in AML cells correlated with a drop in MMP [86]. Quercetin was also shown to induce apoptosis through the activation of the extracellular signal-regulated kinase (ERK) [87,88] in a MAPK (mitogen-activated protein kinase) pathway that intersects with the Akt/mTOR/S6K axis in the regulation of cell survival, proliferation, and cell death [89]. Resveratrol, similar to curcumin, has a high potential for synergistic activity, as it can target multiple pathways responsible for proliferation, cell cycle arrest, apoptosis, and cell growth and survival. It can also induce apoptosis by suppressing ERK signaling and Akt signaling [46,90,91]. Both compounds demonstrated synergistic activity with each other, inhibiting cell growth in the SCC-25 oral squamous carcinoma cell line [92] or increasing MMP in the human pancreatic carcinoma Mia PACA-2 cell line [93]. However, their effect on the viability of tumor cells in vitro could be obtained only at concentrations higher than bioavailable [10,33,36,62,63], as we also observed. While some research showed that their combination has an anti-cancer effect even at low concentrations present in diluted red wine, the presence of other wine phytochemicals was indicated to be responsible for this effect [92].

One of the most significant points of our research was that we demonstrated a difference in the activity of our mix of phytochemicals against normal and neoplastic cells. It did not affect the viability of normal human foreskin fibroblasts (BJ cells). The BJ cell line consists of normal, finite lifespan cells without any dysfunction in homeostasis, constricted in their proliferation by contact inhibition and forming a stable population. While we observed a significant drop in their MMP after 24 h, with prolonged incubation BJ cells, we were able to mostly recover high MMP, even at 2× dose, without any drop in their viability. For the MOLT-4 cell line, while the drop in MMP after the first 24 h was not as steep as the BJ cells, it correlated with a drop in cell viability and inhibition of cell population growth. This difference could be explained by the difference in mitochondrial potential between tumor and normal cells, as it is approximately 60 mV higher in carcinomas as compared to their normal controls. Therefore, selective killing of carcinoma cells can be obtained through mitochondrial toxicity [94]. Furthermore, the significant drop in MMP in BJ cells when the mix of all four nutraceuticals was used can paradoxically block the induction of apoptosis in BJ cells and allow for recovery. In our previous research, we demonstrated that a rapid decrease in MMP might suggest that ATP is depleted and the process of apoptosis might be inhibited, as proper levels of ATP generation are needed for the execution of this process [95]. 

Such decreased levels of ATP due to the reduced MMP were shown to affect the function of Na^+^/K^+^ ATPases and finally contribute to cell swelling and cell membrane permeability (CMP) [96]. Both cell lines demonstrated a similarly sharp increase in CMP at 24 h; however, BJ cells were able to restore their membrane continuity as indicated by viability assays. The ability of tumor cells to migrate and invade requires an increase in cell membrane dynamics. Exposing them to higher physical stress, in association with their altered membrane stiffness, makes them more sensitive to stretch-induced membrane pores/ruptures [97,98]. Even resting MOLT-4 cells have abundant microvilli of various lengths and densities and can undergo polarization [99]. While cell membrane repair is triggered by calcium influx at the injury site [100], mitochondria have an important role as a facilitator of this acute and localized repair response [101,102]. Decreased sensitivity to rupture the cell membrane when pores are formed together with increased ability to restore MMP and thus activity of mitochondria in cell membrane repair allows non-neoplastic cells to be protected against cytotoxic activity of our mix, while the ALL cell line, MOLT-4, still can be affected. 

Many nutraceuticals, especially polyphenols, are investigated in clinical trials, reporting their safety and effectiveness of anti-cancer activity as mixtures of polyphenols and in combination with anti-cancer drugs [103,104]. We have demonstrated here that a novel, complex combination of polyphenols applied at a bioavailable concentration can induce interactive effects and significantly inhibit the viability of MOLT-4 cells without influencing normal human fibroblasts. This indicates that the precise composition of a nutraceutical cocktail could possess antitumor properties with fewer side effects for healthy cells than chemotherapy treatment applied for acute T lymphoblastic leukemia patients. However, further studies conducted on primary cell cultures, animal models, and finally on patients are needed to confirm the results obtained in the presented in vitro research. Nevertheless, our data show that the application of phytochemicals, especially their interactive combinations, is the correct approach for investigating alternative treatments to toxic chemotherapy protocols or for the supplementation of traditional therapy to increase its efficiency and decrease side effects.

Furthermore, a combination of polyphenols can also lead to an improvement in their poor bioavailability in vivo [105]. In particular, quercetin was shown to enhance the bioavailability of other drugs, through pharmacokinetic interaction, as a potent inhibitor of CYP3A4 and a modulator of P-gp [105,106,107]. Its addition to the mix can potentially reduce the activity of curcumin as an inhibitor of the P-gp function [108]. However, the changes in the uptake of the curcumin, genistein, quercetin, and resveratrol mix in vivo still need to be checked experimentally. Furthermore, in designing such an in vivo study, the potential influence of metabolism on the status of phytochemicals in the body after their administration should be carefully considered. Some phytochemicals present in plasma after oral uptake can be metabolized faster in contrast to the slower metabolism of parent compounds after parenteral administration, including genistein and quercetin [43,44].

## 4. Materials and Methods

### 4.1. Cell Lines

An acute T lymphoblastic leukemia cell line, MOLT-4, was acquired from the European Collection of Authenticated Cell Cultures (ECACC, a part of Public Health England, Porton Down, Salisbury, UK) (Cat No. 85011413). BJ, normal human foreskin fibroblasts were purchased from the American Type Culture Collection (ATCC, Manassas, VA, USA) (Cat No. CRL-2522). MOLT-4 cells were maintained long-term as a suspension cell culture in RPMI 1640 medium supplemented with 2 mM Glutamine and 10% FBS (Fetal Bovine Serum). Cells were subcultured by dilution in a fresh medium. BJ cells were maintained short-term as an adherent cell culture in EMEM medium supplemented with 10% FBS and subcultured with a trypsin-EDTA solution. Both cell lines were cultivated in a CO_2_ incubator at 37 °C in a humidified atmosphere containing 5% CO_2_. 

### 4.2. Phytochemicals and Their Combinations

Curcumin from *Curcuma longa* L. (Turmeric, (1*E*,6*E*)-1,7-bis(4-hydroxy-3-methoxyphenyl)hepta-1,6-diene-3,5-dione) (Cat No. C1386), genistein (5,7-dihydroxy-3-(4-hydroxyphenyl)chromen-4-one) (Cat No. G6649), quercetin (2-(3,4-dihydroxyphenyl)-3,5,7-trihydroxychromen-4-one) (Cat No. Q4951), resveratrol (5-[(*E*)-2-(4-hydroxyphenyl)ethenyl]benzene-1,3-diol) (Cat No. R5010) were purchased from Sigma-Aldrich (St Louis, USA) and dissolved in 100% DMSO (dimethyl sulfoxide) to obtain stock solutions (9, 12, 6 and 2 mg/mL, respectively). 

The bioavailable concentrations of phytochemicals chosen for analysis were: 6.1 µM (2.25 µg/mL) for curcumin [53,54], 11.1 µM (3 µg/mL) for genistein [55,56], 4.96 µM (1.5 µg/mL) for quercetin [57,59] and 2.2 µM (0.5 µg/mL) for resveratrol [33,58] (denoted on all graphs as 1×). However, as the range of bioavailability varies in the already published data for each analyzed compound, we also chose to analyze concentrations amounting to half of and double the chosen bioavailable dose (denoted as 0.5× and 2× on all graphs, respectively) (Table 1).

Mixes of phytochemicals were prepared by combining reagents at the same bioavailable concentration in the complete medium supplemented with 10% FBS. At 2× bioavailable concentration, dilution with the medium decreased the DMSO level to 0.05%, below the concentration that can induce biological changes [64,65]. DMSO concentration was adjusted to 0.05% for all subsequent serial dilutions as well as the control.

### 4.3. Cell Viability Assay (MTT Assay)

Cell viabilities of MOLT-4 and BJ cells were assessed with an MTT (3-[4,5-dimethylthiazol-2-yl]-2,5-diphenyltetrazolium bromide) assay [109]. MOLT-4 cells were seeded into the wells of a 96-well plate at a density of 3 × 10^4^ cells per well in 50 μL of the medium, and then serial dilutions of the analyzed reagents were added in 50 μL of the medium to the end volume of 100 μL per well. BJ cells were seeded into the wells of a 96-well plate at a density of 1 × 10^4^ cells per well in 100 μL of the medium, incubated overnight, and then the medium was exchanged for serial dilutions of the analyzed reagents to the end volume of 100 μL per well. The complete medium supplemented with 10% FBS was used. The end concentration of DMSO was kept at the stable level of 0.05%, including the control wells. After 24, 48, and 72 h of incubation in the CO_2_ incubator, cell viability was measured with the MTT assay. Briefly, in this metabolic assay, tetrazolium salt is reduced to purple formazan crystals through the activity of mitochondrial reductase. After 2–3 h, formazan is dissolved with 100 μL of acidified isopropanol and the optical density of the wells is measured at 570 nm. The optical density of the wells containing cells cultured without any analyzed reagents was assumed to represent 100% viability and used as a control. 

### 4.4. Mitochondrial Membrane Potential Assay (MMP Assay)

The assay was based on the method published in Żołnowska et al. [110]. Briefly, MOLT-4 cells were seeded into the wells of a 24-well plate at a density of 2 × 10^5^ cells in 350 µL of the medium, and then serial dilutions of the analyzed reagents were added in 350 μL of the medium to the end volume of 700 μL per well. BJ cells were seeded into the wells of a 24-well plate at a density of 5 × 10^4^ cells in 500 µL of the medium, incubated overnight, and then the medium was exchanged for serial dilutions of the analyzed reagents to the end volume of 700 μL per well. The complete medium supplemented with 10% FBS was used. The end concentration of DMSO was kept at a stable level of 0.05%. The cells were incubated for 24, 48, and 72 h in the CO_2_ incubator, and 30 min before the end of incubation MitoProbe JC-1 was added into each well to the end concentration of 25 µM. After JC-1 staining, the cells were washed with phosphate-buffered saline (PBS), trypsinized (Corning^®^ 25-053CI), resuspended in PBS, and analyzed by flow cytometry at λ excitation (ex) = 488 nm and λ emission (em) = 525/570 nm (LSR II BD Biosciences, New Jersey, NJ, USA). Cells cultured without any analyzed reagents were treated as a control. For a positive control, the cells were treated for 15 min with 200 nM of CCCP (carbonyl cyanide m-chlorophenylhydrazone), a mitochondrial oxidative phosphorylation uncoupler, before JC-1 addition.

### 4.5. Cell Membrane Permeability Assay (CMP Assay)

MOLT-4 and BJ cells were prepared as described for the MMP assay, except that 30 min before the end of incubation, Trypan blue was added into each well to the end concentration of 0.002%, and the cells were analyzed by flow cytometry at excitation λ_ex_ = 488 nm and emission λ_em_ = 610 nm (LSR II BD Biosciences, USA). Cells cultured without any analyzed reagents were treated as a control. The assay was based on the method described in Avelar-Freitas et al. [111]. 

### 4.6. Cell Cycle Analysis

MOLT-4 cells were prepared as described for the MMP assay, except that after cells were incubated for 24, 48, and 72 h with analyzed reagents, cells were fixed in cold 70% ethanol for 30 min. Fixed cells were stained for 15 min in buffer containing 500 µg/mL RNAse A (EURx, Gdansk, Poland) and 2.5 µg/mL of DAPI (Merck, Darmstadt, Germany). The fluorescence of DAPI was measured at excitation λ_ex_ = 355 nm and emission λ_em_ = 440 nm with a flow cytometer (LSR II, BD Biosciences, Mountain View, CA, USA). Each experiment was performed in triplicate. Data were analyzed off-line using the Kaluza Analysis Software 2.1.3 (Beckman Coulter Life Sciences, IN, USA). The assay was based on the method described in Żołnowska et al. [110].

### 4.7. Statistical Analysis

Mean values were obtained from at least three separate experiments with three technical repeats each and reported as the mean (±SD). For the viability assay, Statistica 12 software was used, and the Mann–Whitney test for two unpaired groups of a non-Gaussian population was applied. For the MMP and CMP assays, GraphPad Prism 7.02 software was used, and the 2-way ANOVA test was applied. *p* values < 0.05 were considered significant. The combination index (CI) was analyzed with CompuSyn software.

## 5. Conclusions

We demonstrated that curcumin and genistein at their bioavailable concentration have a significant interactive effect on MOLT-4 cell line viability in vitro. Resveratrol and quercetin did not affect MOLT-4 cell line viability, alone or in a combination, but added together to the genistein/curcumin mix, were able to enhance its anti-cancer effect. The combination of these four polyphenols was non-toxic to normal human fibroblasts. While it induced a decrease in MMP and correlated CMP changes, in non-tumor cells, metabolic activity and cell membrane continuity were restored with time. We can conclude that the “cocktail” of four natural compounds (6.1 µM curcumin, 11.1 µM genistein, 2.2 µM resveratrol, and 4.96 µM quercetin) has a significant interactive anti-cancer effect. Hence, it may be a promising treatment modality to improve the outcomes of acute lymphoblastic leukemia treatment, particularly during the less intensive maintenance therapy that lasts 1.5–3 years, depending on the treatment regimen. Alternatively, introducing a viable anti-cancer “nutraceutical cocktail” upon the conclusion of chemotherapy may also offer an opportunity to reduce the risk of late relapses, and spare pediatric patients from the heavy burden of salvage therapies, while avoiding any risk of non-desirable interactions between these compounds and chemotherapeutic agents. Further research is required to confirm our hypothesis of whether it is applicable in vivo.

## Figures and Tables

**Figure 1 ijms-23-04753-f001:**
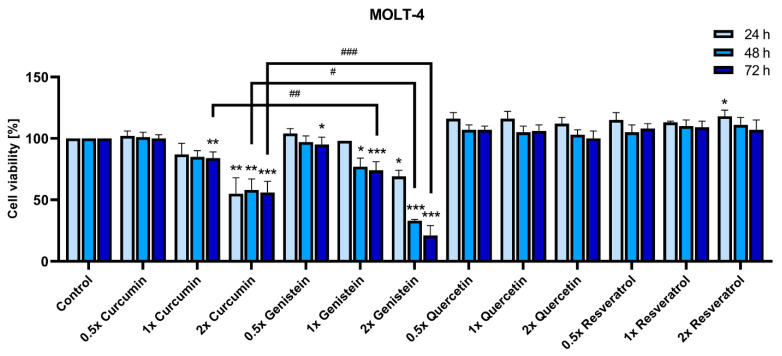
Effect of phytochemicals on MOLT-4 viability according to their bioavailability. Cell viability was determined by MTT metabolic assay at 24, 48, and 72 h after supplementation with 0.5/1/2× bioavailable concentrations specific for each dietary polyphenol: curcumin 3.05/6.1/12.2 µM, genistein 5.55/11.1/22.2 µM, quercetin 2.48/4.96/9.92 µM, or resveratrol 1.1/2.2/4.4 µM, respectively. A non-treated control from each data point was assumed to be 100% viable to demonstrate the relative changes in viability in time at each concentration. The data are presented as the mean ± SD of three independent tests (* *p* < 0.05, ** *p* < 0.01, *** *p* < 0.001 vs. control; ^#^ *p* < 0.05, ^##^ *p* < 0.01, ^###^ *p* < 0.001 between two groups).

**Figure 2 ijms-23-04753-f002:**
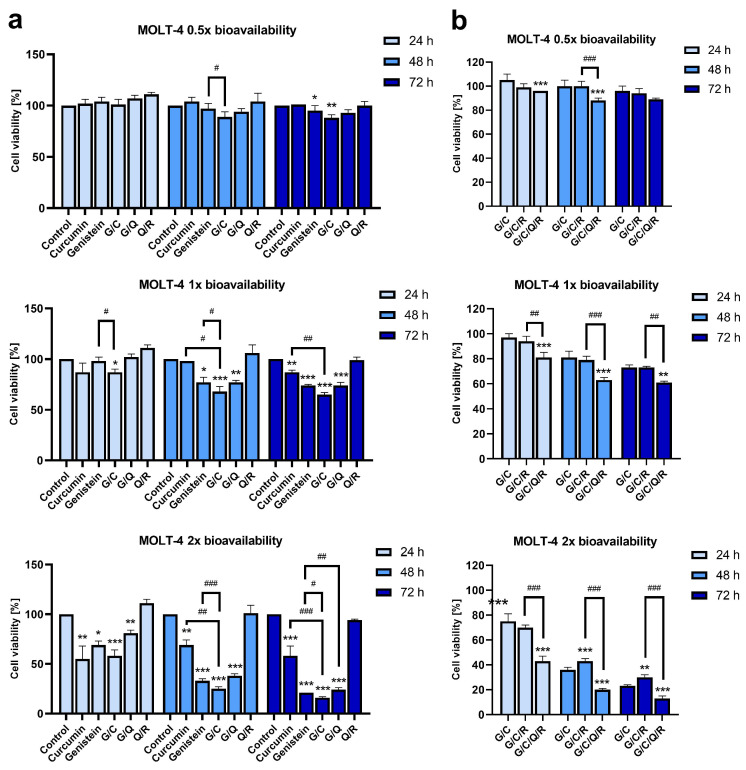
MOLT-4 viability after 24, 48, and 72 h with phytochemical’s combinations according to their bioavailability. Cell viability was determined by MTT metabolic assay at 24, 48, and 72 h after supplementation with different mixes of curcumin (C), genistein (G), quercetin (Q), and resveratrol (R) combined at a comparable dose according to their bioavailability. Bioavailable concentration specific for each dietary polyphenol at 0.5/1/2× dose: curcumin 3.05/6.1/12.2 µM, genistein 5.55/11.1/22.2 µM, quercetin 2.48/4.96/9.92 µM, or resveratrol 1.1/2.2/4.4 µM, respectively. (**a**) Changes in viability induced by curcumin, genistein, and chosen combinations of two polyphenols relative to the non-treated control (100% viability). (**b**) Changes in viability induced by the curcumin and genistein mix and its supplementation with other nutraceuticals. Viability percentages were scored according to the control (100% viability), but the control was not included in the graph. All data points are presented as the mean ± SD of three independent tests (* *p* < 0.05, ** *p* < 0.01, *** *p* < 0.001 vs. control; ^#^ *p* < 0.05, ^##^
*p* < 0.01, ^###^ *p* < 0.001 between two groups).

**Figure 3 ijms-23-04753-f003:**
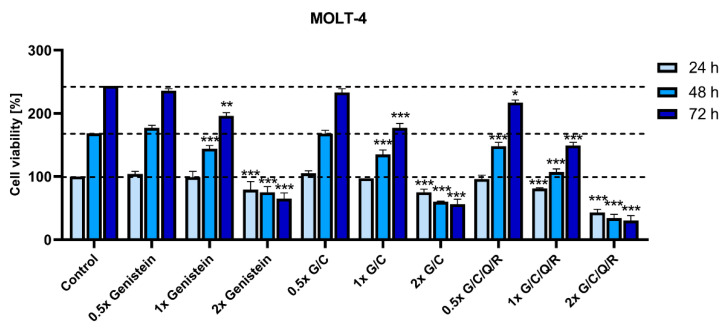
Effect of selected combinations of phytochemicals on MOLT-4 viability in accordance with population growth. Cell viability was determined by MTT metabolic assay at 24, 48, and 72 h after supplementation with genistein, genistein/curcumin mix (G/C), and a mix of all analyzed polyphenols (G/C/Q/R). The viability of a non-treated control at 24 h was treated as 100% viability, and the other percentages were scored accordingly. Polyphenols were combined at a comparable dose according to their bioavailable concentration at 0.5/1/2× dose: curcumin 3.05/6.1/12.2 µM, genistein 5.55/11.1/22.2 µM, quercetin 2.48/4.96/9.92 µM, or resveratrol 1.1/2.2/4.4 µM, respectively. The data are presented as the mean ± SD of three independent tests (* *p* < 0.05, ** *p* < 0.01, *** *p* < 0.001 vs. respective control).

**Figure 4 ijms-23-04753-f004:**
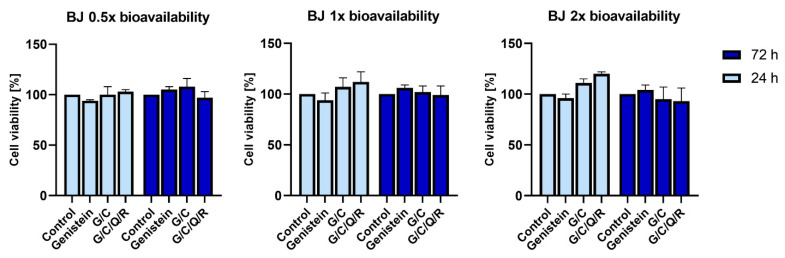
BJ viability after 24 and 72 h with phytochemical combinations according to their bioavailability. Cell viability was determined by MTT metabolic assay at 24 and 72 h after supplementation with genistein, genistein/curcumin mix (G/C), and a mix of all analyzed polyphenols (G/C/Q/R). A non-treated control from each data point was assumed to be 100% viability to demonstrate the relative changes in viability in time at each concentration. Polyphenols were combined at a comparable dose according to their bioavailable concentration at 0.5/1/2× dose: curcumin 3.05/6.1/12.2 µM, genistein 5.55/11.1/22.2 µM, quercetin 2.48/4.96/9.92 µM, or resveratrol 1.1/2.2/4.4 µM, respectively. The data are presented as the mean ± SD of three independent tests. Observed differences were not statistically significant.

**Figure 5 ijms-23-04753-f005:**
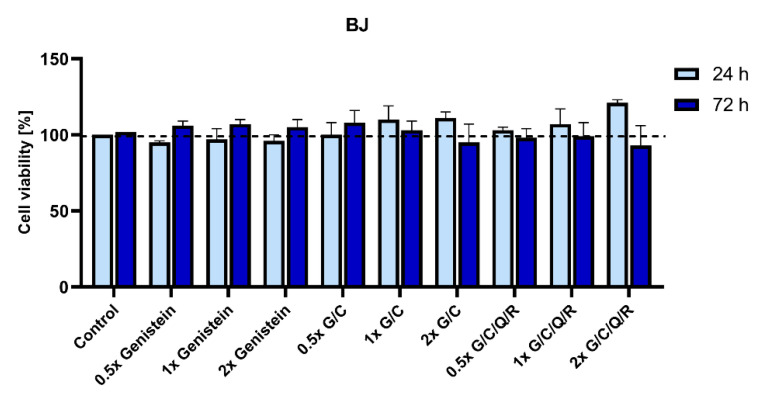
Effect of selected combinations of phytochemicals on BJ viability in accordance with population growth. Cell viability was determined by MTT metabolic assay at 24 and 72 h after supplementation with genistein, genistein/curcumin mix (G/C), and a mix of all analyzed polyphenols (G/C/Q/R). The viability of a non-treated control at 24 h was treated as 100% viability, and the other percentages were scored accordingly. Polyphenols were combined at a comparable dose according to their bioavailable concentration at 0.5/1/2× dose: curcumin 3.05/6.1/12.2 µM, genistein 5.55/11.1/22.2 µM, quercetin 2.48/4.96/9.92 µM, or resveratrol 1.1/2.2/4.4 µM, respectively. The data are presented as the mean ± SD of three independent tests. Observed differences were not statistically significant.

**Figure 6 ijms-23-04753-f006:**
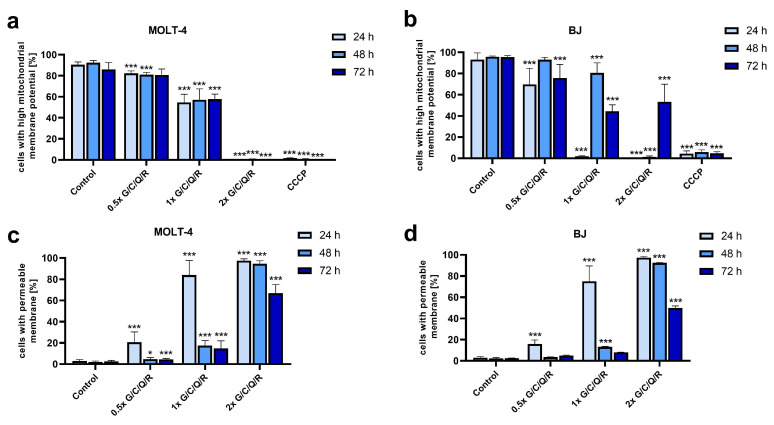
Changes in mitochondrial membrane potential and cell membrane permeability induced by a combination of phytochemicals. Cells were treated for 24, 48, and 72 h with a combination of genistein/curcumin/quercetin/resveratrol (G/C/Q/R) combined at a comparable dose according to their bioavailable concentration at 0.5/1/2× dose: curcumin 3.05/6.1/12.2 µM, genistein 5.55/11.1/22.2 µM, quercetin 2.48/4.96/9.92 µM, or resveratrol 1.1/2.2/4.4 µM, respectively. (**a**,**b**) Mitochondrial membrane potential (MMP) of MOLT-4 (**a**) and BJ cells (**b**) was measured with non-treated cells as a negative control, and CCCP-treated cells as a positive control for low MMP. (**c**,**d**) Cell membrane permeability (CMP) of MOLT-4 (**c**) and BJ cells (**d**) was measured with non-treated cells as a negative control. The data are presented as the mean ± SD of three independent tests (* *p* < 0.05, *** *p* < 0.001 vs. respective control).

**Table 1 ijms-23-04753-t001:** Concentrations of each nutraceutical used and their correlation with their bioavailability in vivo.

	Bioavailability		
	0.5×	1×	2×
Curcumin	3.05 µM	6.1 µM (2.25 µg/mL) [53,54]	12.2 µM
Genistein	5.55 µM	11.1 µM (3 µg/mL) [55,56]	22.2 µM
Quercetin	2.48 µM	4.96 µM (1.5 µg/mL) [57,58]	9.92 µM
Resveratrol	1.1 µM	2.2 µM (0.5 µg/mL) [33,59]	4.4 µM

## Data Availability

All data generated or analyzed during this study are included in this published article and its Appendix A.

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
