# Peer review of "The Cooperative Anti-Neoplastic Activity of Polyphenolic Phytochemicals on Human T-Cell Acute Lymphoblastic Leukemia Cell Line MOLT-4 In Vitro"

_ijms, 2022, doi:10.3390/ijms23094753_

Round 1

Reviewer 1 Report

This manuscript investigated the cooperative effect of phytochemicals (curcumin, genistein, quercetin, and resveratrol) at physicals doses on the cell viability of human T-cell acute lymphoblastic leukemia cell line and the possible mechanism. This study is interesting.

The major concern:

The authors used the parent form of the phytochemicals to perform the in vitro study. However, the major compounds of these phytochemicals present in plasma after oral uptake are not these parent forms because of the quick metabolism of these phytochemicals by phase II enzymes. In contrast, these parent compounds exit in plasma after iv injection. These should be mentioned in the discussion.

The minor concerns  

1. the legend of figure 1 (not 2) is incorrect.

2. ine 334-340: “We have observed that while a combination of all four polyphenols at a 1x bioavailable dose (6.1 μM curcumin, 11.1 μM genistein, 4.96 μM quercetin, and 2.2 μM resveratrol) was unable to significantly reduce the tumor cell population below the starting point (it required 2x dose applicable only in vitro) it was able to suppress its growth, especially during the first 48 hours of treatment(Figure 3)”. This paragraph is confusing because according to fig. 3, 1x G/C/Q/R significantly decreased cell viability compared to the control at the same time point. Although the cell population still increased the number is significantly lower than in the control group.   

Author Response

Responses to the Reviewers’ comments

Reviewer No. 1

This manuscript investigated the cooperative effect of phytochemicals (curcumin, genistein, quercetin, and resveratrol) at physicals doses on the cell viability of human T-cell acute lymphoblastic leukemia cell line and the possible mechanism. This study is interesting.

We appreciate the kind interest of the Reviewer in our work.

1) The major concern:

The authors used the parent form of the phytochemicals to perform the in vitro study. However, the major compounds of these phytochemicals present in plasma after oral uptake are not these parent forms because of the quick metabolism of these phytochemicals by phase II enzymes. In contrast, these parent compounds exit in plasma after iv injection. These should be mentioned in the discussion.

Thank you very much for your suggestion. Indeed, the aspect of metabolism is the issue that could be considered in the research program designed for the discoveries on the application of phytochemicals. We have added, at the end of the Discussion section, short information on the potential influence of metabolism on the status of phytochemicals in the body after their administration. One of the studies referenced here is a review of the bioavailability and pharmacokinetics of genistein, where this problem is very clearly demonstrated. This is a good point to consider before the in vivo research, our next step.

2) The minor concerns  

  1. the legend of figure 1 (not 2) is incorrect.

Thank you for the correction. We have modified the incorrect numbering of Figure 1.

  1. line 334-340: “We have observed that while a combination of all four polyphenols at a 1x bioavailable dose (6.1 μM curcumin, 11.1 μM genistein, 4.96 μM quercetin, and 2.2 μM resveratrol) was unable to significantly reduce the tumor cell population below the starting point (it required 2x dose applicable only in vitro) it was able to suppress its growth, especially during the first 48 hours of treatment (Figure 3)”. This paragraph is confusing because according to fig. 3, 1x G/C/Q/R significantly decreased cell viability compared to the control at the same time point. Although the cell population still increased the number is significantly lower than in the control group.   

Thank you for pointing this out. We modified the text to better express our meaning, that the cell population was still growing when 1x mix was used, unlike in the case of 2x mix, when the cell population was decreasing over time. We hope it is now clearer.

Reviewer 2 Report

Present work discuss about the polyphenolic phytochemical for the treatment of lymphoblastic leukemia treatment. Few points will be useful

  1. Abstract should be unstructured
  2. Introduction Line 126 to 133. It is result part, I don't think it should be part of introduction. 
  3. Authors just tested antileukemia potential of the polyphenol combinations. Authors have simply quoted flow cytometry data from previous study, and assumed that it will be similar situation in their combination strategy. Authors are requested to provide flow cytometry data and for their combination treatment as well for having idea about the cycle cycle arrest.
  4. Conclusion: Line:575 should be considered as limitation or create separate subsection of future direction.
  5. Please provide specific conclusion on what concentration and combination for polyphenol works best for leukemia treatment. In current format conclusion is like suggestion for future investigation instead of concluding their own study. 

Author Response

Responses to the Reviewers’ comments

Reviewer No. 2

Present work discusses about the polyphenolic phytochemical for the treatment of lymphoblastic leukemia treatment. Few points will be useful

Thank you for your recommendations.

Abstract should be unstructured

The construction of Abstract is prepared according to the journal editorial rules.

Introduction Line 126 to 133. It is result part, I don't think it should be part of introduction.

We have deleted the description of our results from the Introduction section.

Authors just tested antileukemia potential of the polyphenol combinations. Authors have simply quoted flow cytometry data from previous study, and assumed that it will be similar situation in their combination strategy. Authors are requested to provide flow cytometry data and for their combination treatment as well for having idea about the cycle arrest.

Thank you for your comment regarding the cell cycle analysis in cells treated with analyzed nutraceuticals. Our preliminary experiments excluded arrest in the cell cycle (cytostatic effects of nutraceuticals) on MOLT-4. However, your valuable comment convinced us to add the information on cell cycle analysis in the text of the manuscript. We have added Fig. 3 to the Supplementary Material, demonstrating the lack of any statistically significant differences in the cell cycle between control cells and cells treated with nutraceuticals. We have also added the explanation in the Results section and modified the Methods and Materials section.

Conclusion: Line:575 should be considered as limitation or create separate subsection of future direction.

Please provide specific conclusion on what concentration and combination for polyphenol works best for leukemia treatment. In current format conclusion is like suggestion for future investigation instead of concluding their own study.

We have modified the Conclusions section according to the kind recommendations of the Reviewer. Thank you for the improvement of our manuscript.

Reviewer 3 Report

The paper submitted by Koszalka et al. investigates the in vitro effects of four polyphenols (alone or in different combinations) using both fibroblasts and leukemia cell lines. 

The idea is very interesting, the manuscript is clear, well written and the conclusions are supported by the results. However, some minor corrections are needed before its publication:

  1. in the introduction section, the authors must discuss about the different polymer-based systems used for the encapsulation of polyphenols in order to increase their bioavailability and solubility. some useful references can be: https://doi.org/10.3390/ijms22063075; https://doi.org/10.3390/polym12071450; https://doi.org/10.1016/j.giant.2020.100022; http://dx.doi.org/10.3390/molecules25204613 
  2. no results should be provided in the introduction section therefore please delete the last paragraph.
  3. in the conclusion section must be provided more detailed results and not only generalities. please indicate which is the best combination and which product has not an anti-neoplastic activity. 

Author Response

Responses to the Reviewers’ comments

Reviewer No. 3

The idea is very interesting, the manuscript is clear, well written and the conclusions are supported by the results. However, some minor corrections are needed before its publication.

We appreciate the kind interest of the Reviewer in our study.

  1. In the introduction section, the authors must discuss about the different polymer-based systems used for the encapsulation of polyphenols in order to increase their bioavailability and solubility. some useful references can be: https://doi.org/10.3390/ijms22063075https://doi.org/10.3390/polym12071450https://doi.org/10.1016/j.giant.2020.100022http://dx.doi.org/10.3390/molecules25204613

It is a good point. Of course, a more straightforward discussion in that field would require additional study and significantly increase the length of the Introduction section. However, we have mentioned this issue in our Introduction to indicate that point for the significant consideration by the potential readers of the final paper. Thank you for this suggestion.

  1. No results should be provided in the introduction section therefore please delete the last paragraph.

We have deleted the description of our results from the Introduction section.

  1. In the conclusion section must be provided more detailed results and not only generalities. please indicate which is the best combination and which product has not an anti-neoplastic activity. 

Thank you for pointing this out. We added a more detailed description of our results in this section and pointed out the best combination and which product has not an anti-neoplastic activity. We hope it is now clearly stated.

Round 2

Reviewer 2 Report

I think authors have answered all the questions raised and it can be accepted